# GPU accelerated atmospheric chemical kinetics in the ECHAM/MESSy (EMAC) Earth system model (version 2.52)

Michail Alvanos[1] and Theodoros Christoudias[1]

[1]*The Cyprus Institute*, PO Box 27456, 1645 Nicosia, Cyprus

*Correspondence to:* Theodoros Christoudias (christoudias@cyi.ac.cy)

**Abstract.**

This paper presents an application of GPU accelerators in Earth system modelling. We focus on atmospheric chemical kinetics, one of the most computationally intensive tasks in climate-chemistry model simulations. We developed a software package that automatically generates CUDA kernels to numerically integrate atmospheric chemical kinetics in the global climate model ECHAM/MESSy Atmospheric Chemistry (EMAC), used to study climate change and air quality scenarios. A source-to-source compiler outputs a CUDA compatible kernel, by parsing the FORTRAN code generated by the Kinetic Pre-Processor (KPP) general analysis tool. All Rosenbrock methods that are available in the KPP numerical library are supported.

Performance evaluation, using Fermi and Pascal CUDA-enabled GPU accelerators shows achieved speedups of $4.5\times$ and $20.4\times$ respectively of the kernel execution time. A node-to-node real-world production performance comparison shows a $1.75\times$ speed-up over the non-accelerated application using the KPP 3-stage Rosenbrock solver. We provide a detailed description of the code optimizations used to improve the performance including memory optimizations, control code simplification, and reduction of idle time. The accuracy and correctness of the accelerated implementation are evaluated by comparing to the CPU-only code of the application. The median relative difference is found to be less than 0.000000001% when comparing the output of the accelerated kernel the CPU-only code.

The approach followed, including the computational workload division and the developed GPU solver code can potentially be used as the basis for hardware acceleration of numerous geoscientific models that rely on KPP for atmospheric chemical kinetics applications.

## 1 Introduction

One of today's great scientific challenges is to predict how climate will change locally as gas concentrations change over time. The study of chemistry-climate interactions represents an important and, at the same time, difficult task of global Earth system research. The emerging issues of climate change, ozone depletion and air quality, which are challenging from both scientific and policy perspectives are represented in Chemistry-Climate Models (CCMs). Understanding how the chemistry and composition of the atmosphere may change over the 21st century is essential in preparing adaptive responses or establishing mitigation strategies.

The global Atmosphere-Chemistry model ECHAM/MESSy (EMAC) is a numerical chemistry and climate simulation system that includes sub-models describing tropospheric and middle atmosphere processes and their interaction with oceans, land and human influences (Jöckel et al., 2010). It uses the second version of the Modular Earth Submodel System (MESSy2) to link multi-institutional computer codes. The core atmospheric model is the 5th generation European Centre Hamburg general circulation model (Roeckner et al., 2006). The EMAC model runs on several platforms, but it is currently unsuitable for massively parallel computers, due to its scalability limitations. In climate simulation applications, the numerical integration by chemical kinetics solvers can take up to 90% of execution time (Christou et al., 2016). To achieve realistic simulation times, researchers are forced to limit the resolution of model simulations.

This paper describes a method of accelerating the chemical kinetics calculations on modern high-performance supercomputers using GPU accelerators. We present a source-to-source parser, written in the Python programming language, that transforms the MESSy chemical kinetics FORTRAN source code to CUDA source code, suited for running on CUDA-enabled general purpose graphics processing unit (GPGPU) accelerators. The parser transforms the auto-generated FORTRAN code by the KPP preprocessor (Sandu and Sander, 2006; Damian et al., 2002b) into the CUDA-compatible accelerated code, allowing to offload all different numerical integration solvers to GPU accelerators. The parser also makes the appropriate changes in the MESSy software distribution for linking the accelerated code during the compilation phase.

The paper is organized as follows: Sec. 1.1 describes the Messy/MECCA frameworks, the parallelization approaches, and Sec. 1.2 the potential of GPU accelerators. Sec. 1.3 discusses previous work that is related to this research. In Sec. 2, we present our implementation of atmospheric chemical kinetics parallelisation, the source-to-source parser code, and GPU-specific optimizations, including memory optimizations, the control code restructuring, and the refactoring of the EMAC source code. An evaluation of the resulting GPU accelerated climate model appear in Sec. 3. We summarize the main outcomes, present our conclusions and planned future work in Sec. 4.

## 1.1 The EMAC framework

The numerical global atmosphere-chemistry model EMAC (ECHAM/MESSy Atmospheric Chemistry) is a modular global model that simulates the chemistry and dynamics of the stratosphere and troposphere. The model includes different sub-models for the calculation of concentrations in the atmosphere, their interaction with the ocean and land surfaces, and the anthropogenic influences. The EMAC model runs on several platforms, but it is currently unsuitable for massively parallel computers, due to its scalability limitations and large memory requirements per core.

The MESSy submodel MECCA executes independently the gas phase chemical kinetics because there are no dependencies between physical neighbours and no limitations by vertical adjacency relations. In a typical configuration of MESSy with 155 species and 310 chemical reactions, MECCA takes 70% of the simulation execution time (Christou et al., 2016). The percentage of execution time can go up to 90% in simulations with more complex chemistry.

Currently, EMAC uses coarse-grained parallelism based on the Message Passing Interface (MPI). However, the current approach does not benefit from the accelerators that exist in modern hybrid HPC architectures. This puts severe limitations on

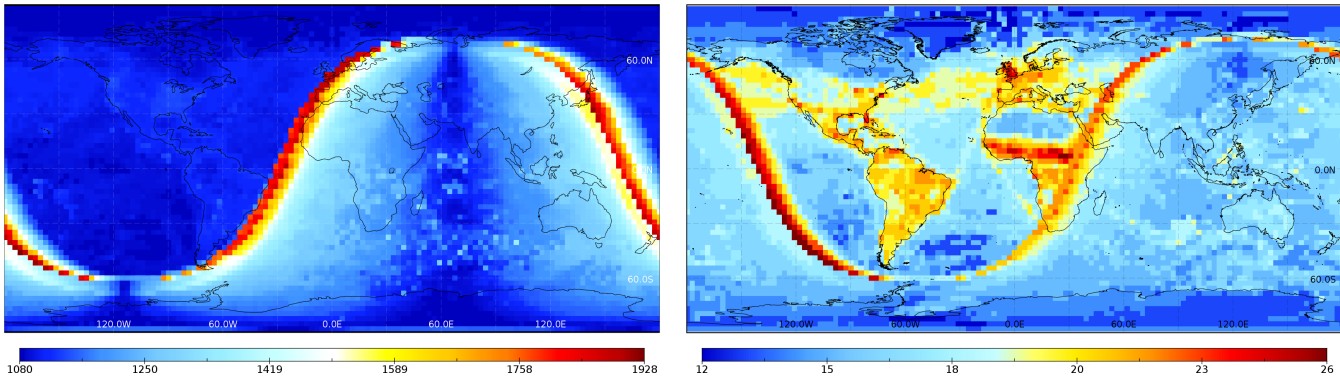

**Figure 1.** Number of integration steps for column MECCA kernel execution (left), and Intra-column maximum difference in execution steps (right). The adaptive time-step integrator shows a non-uniform run time caused by stratospheric photochemistry and natural and anthropogenic emissions.

current global climate time-length atmospheric chemistry and pollution transport simulations in terms of portability, complexity, and resolution.

EMAC uses the Kinetic Pre-processor (KPP) (Sandu and Sander, 2006; Damian et al., 2002b) open source general analysis tool to formulate the chemical mechanism. KPP integrates very efficient numerical analysis routines and automatically generates FORTRAN and C code that computes the time evolution of chemical species from a specification of the chemical mechanism in a domain-specific language. Taking a set of chemical reactions and their rate coefficients as input, KPP generates code of the resulting coupled ordinary differential equations (ODEs). Solving the ODE system allows the temporal integration of the kinetic system. Efficiency is obtained by exploiting the sparsity structures of the Jacobian matrix.

The biggest challenge to address in the application is the imbalance caused by the adaptive time-step integrator solving the differential equations that describe the chemical equations computed. The varying light intensity at sunrise and sunset in combination with concentrations of precursors and gasses (such as $NO_x$ and $O_3$) lead to photochemical reactions (mostly over mid-latitudes in the stratosphere) that heavily alter the stiffness of the ODEs. For example, Fig. 1(left) presents the cumulative number of execution steps required for the integration process for each model column and Fig. 1(right) presents the maximum difference in the number of steps between cells in each column. The difference in the number of steps inside and between columns provides an indication of the strong imbalance created between execution times of different processes.

The ECHAM atmospheric dynamical circulation phase of EMAC only scales up to approximately a few hundred cores (Christou et al., 2016), due to the heavy all-to-all communication overhead of the spectral decomposition. At higher levels of parallelism, at or beyond approximately 1000 cores, the MECCA load imbalance due to the photochemistry also becomes a limiting factor.

## 1.2   GPU Accelerators

A common trend in computing today is the utilization of hardware accelerators that efficiently execute codes rich in data parallelism, to form high-performance heterogeneous systems. Graphical Processing Units (GPUs) are commonly used as accelerators due to high peak performance offered. GPU accelerators are module extensions connected using an interconnect to the CPU and memory sub-system. GPU accelerators feature on-board memory that provides high-bandwidth, medium-latency access to data. In contrast with the general purpose CPUs, the GPU processors are designed for massive parallelism by issuing thousands of simple parallel instructions. Programming a GPU accelerator can be a hard and error-prone process that requires specially designed programming models, such as the CUDA (Nvidia, 2015b) and OpenCL (Munshi, 2009). To unlock the potential performance of accelerators, programmers often apply incremental optimizations to their applications.

## 1.3   Related Developments

There are numerous efforts documented in the literature to improve the performance of climate-chemistry model simulations, specifically targeting chemical kinetics.

The adoption of intrinsic optimizations for sparse linear algebra (Zhang et al., 2011; Jacobson and Turco, 1994) is one of the first improvements implemented to speed up simulations of chemical kinetics. In a chemical kinetic solver, the majority of the time is spent solving the linear systems using implicit integration mechanisms. Only a fraction of the matrices have values different than zero, allowing for more efficient implementations using the sparsity of the structures. The sparse structure depends on the chemical reactions and thus the linear algebra can be solved off-line before the execution of the application. In a typical chemical mechanism, the pattern of chemical interactions leads to a sparse Jacobian with the majority of entries equal to zero. The KPP uses sparse linear algebra to reduce the execution time. The sparsity structure depends only on the chemical network and not on the values of concentrations or rate coefficients. Thus, the lookup tables are constant, allowing to KPP to unroll the loops and remove the indirect memory accesses.

Researchers have also improved the performance of the chemical kinetics portions of the code by using data and instruction level parallelism (DLP and ILP) when solving the chemical kinetics equations. Approaches include the introduction of SIMD instructions and annotation using the OpenMP programming model. However, these approaches often rely on the ability of the compiler to auto-vectorize the code that very often misses opportunities (Zhang et al., 2011), and the OpenMP implementation heavily burdens the performance due to high overhead of scheduling and managing threads.

A more coarse-grained approach is to use grid- or box- level parallelization (Linford et al., 2009; Linford, 2010; Christoudias and Alvanos, 2016). The application breaks the grid or box in cells allowing the calculation of concentrations independently between cells. Thus, the application assigns an equal number of cells to each thread to allow embarrassingly parallel execution of the chemical solvers. The biggest drawback of this approach is the limited parallelism and the imbalance that is created due to the photochemical processes in the lower stratosphere. A similar approach is used for the parallelization of chemical kinetics in the Cell processor and GPU accelerators by Damian et al. (2002a). However, these approaches exhibit the same limitations: limited parallelism, and imbalance. Moreover, it is necessary for researchers to effectively rewrite their applications every time

they run on the accelerators. Current commercial approaches of parallelizing the chemical kinetics use fine-grained parallelism, that is suitable only when the number of elements and chemical reactions are complex enough to justify the use of accelerators.

A different organization of the workload using Adaptive Mesh Refinement (AMR) was proposed as a way to increase locally the resolution for an area of interest (Keppens et al., 2003). The adaptive unstructured mesh representation in climate models can improve the overall performance of the application (Schepke and Maillard, 2012; Schepke, 2014). Compared to traditional horizontal partitioning these solutions offer greater scalability. However, the communication demands are dominant in a high number of processes and the potential imbalance created by the chemical reaction and the heterogeneity of the modern HPC machines are ignored by this approach.

An earlier prototype of the application in this paper is outlined in Christoudias and Alvanos (2016), focusing on the challenges of using GPU accelerators to exploit node-level heterogeneity. This paper significantly expands on the previous work, both in detailed implementation and optimization. Moreover, this paper presents a performance evaluation of the first public release of the source code (Alvanos and Christoudias, 2017a).

## 2   Implementation

The section presents the implementation of the source-to-source parser and the challenges addressed. The parser is written in the Python programming language and generates CUDA (Nvidia, 2015b) compatible solvers, by parsing the auto-generated FORTRAN code output by the KPP preprocessor. This approach avoids the distribution of additional compiler framework, such as the ROSE compiler framework (Quinlan and Liao, 2011; Quinlan et al., 2012), that may contain additional software dependencies under a different license, and at the same time allows easy distribution and maintainability. The process is automated and allows the parser to work with all possible user-defined chemical mechanisms, without requiring changes by the end-user on the accelerated kernel code. The FORTRAN compiler links the output CUDA object file with the rest of the application.

The user executes the parser from the `messy/util` directory to transform the code. The parser modifies the `messy/smcl/messy_mecca_kpp.f90` file and places a single call to the CUDA source file that contains the accelerated code (`messy/smcl/messy_mecca_kpp_acc.cu`) and a wrapper function for issuing the parallel kernels and copying the data to and from the GPU. The solver supports all five variations of Rosenbrock solvers available in KPP (Ros2,Ros3,Ros4,Rodas3, and Rodas4). The parser is also responsible for making the changes to the EMAC source Makefile for linking of object files during compilation. The user must modify the configuration of the EMAC model to include the CUDA runtime during linking phase. Similar to the FORTRAN implementation, the computation is subdivided in runtime-specified arrays of columns. The memory of each array transferred to the GPU global memory and each grid box calculated on individual GPU threads.

The CUDA chemical kinetics solver comprises three steps, also presented diagrammatically in Figure 2. Each task is offloaded using three different calls:

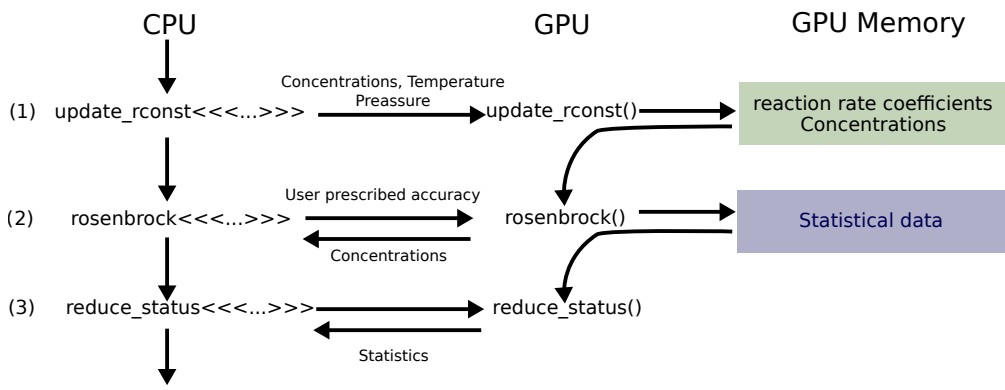

**Figure 2.** Flow-chart of the tasks offload execution on GPU accelerators.

1. The first step is the calculation of the reaction rate coefficients. The variable values are stored in a global array inside the GPU and used in the second step.

2. The second step is the ODE solver that includes all linear algebra functions. The computational kernel contains five optional variations of the Rosenbrock family of numerical solvers, specified by the user at runtime – as supported by KPP/KP4.

3. The third and final step of the solver is the statistical reduction of the results, which demands very limited computational time compared with the other steps.

---

$\text{kpp\_integrate}(time\_step\_len, Conc, gstatus, VL\_GLO)$

1: // $VL\_GLO$ is NVL × levels_atmosphere
2: **for** DO $k = 1$ to $VL\_GLO$ **do**
3:     $status \leftarrow \emptyset$;
4:     $C(:) \leftarrow Conc(k,:)$
5:     $update\_rconst(C)$;
6:     $integrate(\ time\_step\_len, C, status)$        // Main kernel
7:     $Conc(k,:) \leftarrow C(:)$
**Diagnostic Information**
8:     $gstatus(k) \leftarrow status$
9: **end for**

Algorithm 1: CPU Chemical Kinetics Solver.

The parser replaces the code of integrator loops located in the `messy_mecca_kpp.f90` file with a single call to the CUDA accelerated code. The CUDA source file includes an entry function that is responsible for moving the data to and from the GPU accelerator and issuing the computation kernels. Algorithm 1 presents the outline of the integrator code and Algorithm 2

kpp_integrate_acc($time\_step\_len, Conc, gstatus, VL\_GLO$)

1: $status \leftarrow \emptyset$;

**Step 1: Initialize devices and allocate memory.**

2: **if** $initialized$ is $FALSE$ **then**

3:     $dev\_conc\_ptr, dev\_rconst\_ptr, dev\_status\_ptr \leftarrow init\_first\_time(\,pe, VL\_GLO)$;

4: **end if**

5: $cudaMemcpy(\,Conc(:), device\_conc\_ptr, sizeof(double) \times VL\_GLO \times NSPEC\,)$

6: $Blocking \leftarrow 64$;        // Blocking can be changed to 128 by modifying the produced file

**Step 2: Call kernels.**

7: $update\_rconst <<< VL\_GLO/Blocking, Blocking >>> (device\_conc\_ptr, device\_rconst\_ptr, VL\_GLO)$;

8: $Rosenbrock <<< VL\_GLO/Blocking, Blocking >>> (device\_conc\_ptr, device\_rconst\_ptr, dev\_status\_ptr, VL\_GLO)$;

9: $reduce\_status <<< VL\_GLO/Blocking, Blocking >>> (\,dev\_status\_ptr, VL\_GLO)$;

**Step 3: Copy back the results.**

10: $cudaMemcpy(\,device\_conc\_ptr, Conc(:), sizeof(double) \times VL\_GLO \times NSPEC\,)$

11: $cudaMemcpy(\,dev\_status\_ptr, gstatus, sizeof(double) \times VL\_GLO\,)$

Algorithm 2: GPU Accelerated Chemical Kinetics Solver.

presents the outline of the GPU glue code. Each GPU thread calculates the chemical concentrations of an individual cell. The temporary arrays used between steps of the solver are allocated in the stack memory with the exception of the RCONST array. The RCONST array contains the reaction rate coefficients that are re-calculated in the first step of the integration process, or at each substep of the numerical solver. The developed source-to-source parser automatically recognizes the coefficients that

are re-calculated in the first step of the integration process, or at each substep of the numerical solver for an ODE system with varying (rate) coefficients, which has to be integrated as a non-homogeneous system.

The GPU solvers allocate stack memory for the intermediate results revealing good performance in the current and forthcoming generations of GPU architectures due to simpler indexing. However, the design architecture for memory allocation is subject to change when the impact of coalesced memory access is more prominent in in future GPU generations. During the

application development, we discovered that a large number of registers causes spillage to stack memory. Thus, the impact of memory coalescing is limited due to stack occupancy by each CUDA thread. If the number of available registers increases or the local memory becomes larger, then the performance gain from coalescing memory access can be significant. The trend in the GPU architectures is the increase of the on-chip local memory.

The global GPU memory size suffices for offloading the full set of data, and it is not a limiting factor for the present and

future projected nominal chemistry mechanism complexity. The performance is also not limited by the overuse of any function unit. Global and local memory is used to store all required data, and local variables are used for each grid box for temporary variables during computation, with cache misses and register usage being the current performance limiting factor. The block size of the GPU kernels is set to 64, and it can be changed to 128 for increased efficiency for future GPU architectures if

applicable. The maximum number of cells that can be offloaded to the GPU is 12288. This sets an upper limit to the value of the ECHAM5 `NVL` runtime parameter at 128, assuming 90 levels for the atmosphere.

The accelerated CPU process requires a chunk of the GPU VRAM memory, whose size is dependent on the number of species and reaction constants in the MECCA mechanism. The required stack memory can be calculated as follows: $stack\_allocated\_memory = (number\_of\_elements \times 13 + lu\_nonzero \times 2 + spilledvariables) \times sizeof(double)$. For example, in a workload with 155 species and 310 reactions, the required stacked memory including the temporary allocate arrays is $(155 \times 13 + 2 \times 1500 + 1200) \times 8 = 40120$ approximately. Note that the size of spilled variables depends on the complexity of the source code. The total memory required for a kernel is calculated for 12288 cells is: $12288 \times (stack\_allocated\_memory + global\_allocated\_mem) = 12288 \times 52000 \approx 638MB$. Moreover, during the GPU code loading, the GPU driver unrolls the loops and expands the offloaded code making it a limiting factor for allocated memory. For instance, the aforementioned chemistry workload requires at least 2GB of VRAM to be available on the Fermi architecture or more than 6GB in the newest Pascal architecture. We believe that the additional allocated memory is due to the driver side allocation of additional caches and buffers to achieve the best performance. Furthermore, to support the newest features, such as the Unified Virtual Memory model, additional memory allocation is required. Thus, the total memory required for each process depends not only on the size of chemistry but also in the complexity of the source code (number of reactions) and the architecture of the GPU accelerator.

## 2.1 Challenges and Optimizations

To achieve the best possible performance and efficiency three challenges had to be addressed: First, the biggest limitation in the performance of the chemical kinetics kernel is the memory required. To solve the ODEs and calculate the new concentrations, more memory is required in total than the available on-chip memory. For instance, a set of 155 species and 310 reactions requires the use of ~50KB of stack memory per cell. Thus, running multiple threads concurrently on the GPU forces the use of global memory for storing the data. Second, the source code complexity of the solver kernel increases the usage of registers and limits the GPU Streaming Multiprocessor (SM) utilization. Third, the number of steps for solving the differential equations differs between different cells. This creates thread divergence, limiting the performance.

To address these challenges we implemented incremental improvements in the source code running on the GPUs. Our aim is to improve the GPU occupancy, reduce the memory impact, simplify the source code, and reduce the idle time.

### 2.1.1 Occupancy improvement

The high register pressure limits the occupancy, the number of concurrent threads per Streaming Multiprocessor (SM), and increases the overall execution time of the kernel. There are two ways of overcoming this challenge: either place a flag during compilation that limits the number of registers or use the Launch Bound qualifier (`__launch_bounds__`) to a specific kernel. Our implementation uses the second option as a generic, future-proof approach. The downside of limiting the register usage is the increase of register spilling, causing an increase of stack memory usage. The compiler allocates additional stack memory for spill loads and stores, creating additional local memory traffic, that does not fit into the on-chip memory. Thus, the application execution time is dominated by the global memory access latency.

### 2.1.2 Memory optimizations

GPU accelerators employ a wide memory bus that allows for high-bandwidth transfers at the cost of high latency. To achieve the best execution efficiency on the GPU, we apply a number of memory optimizations: i) better utilization of the on-chip memory, ii) privatization of each thread data structure, and iii) prefetching.

Each SM contains a small amount of local memory that can be used as shared memory between threads or for Level-1 (L1) cache. The size of temporary matrices is larger than the available on-chip memory. Thus, only a small portion of the memory can be used to store the data. In particular, the concentrations of chemical species that remain unchanged by chemical kinetics are stored in this memory. To increase the utilization of the local memory, we increased the portion of the L1 cache against the shared memory, through the `cudaFuncCachePreferL1` runtime call.

The solver must keep intermediate results in temporary arrays during different steps of the solver. The temporary arrays are larger than the available on-chip memory, forcing us to declare the arrays in global memory. Although accesses on these arrays can be coalesced, there is still overhead occurring due to cache misses. Moreover, different execution paths of the kernel can complicate and limit the coalescing of the data. To overcome this, we privatized the majority of the matrices, by either replacing them with scalar variables or by using stack-allocated arrays, allowing simplified temporary array indexing code.

The last modification of the memory optimizations is to employ prefetching of data. Prefetching may lead to unpredictable behavior, especially in massively parallel architectures, such as GPUs. There are four possible prefetch instructions that can be applied to the chemical kinetics. Micro-benchmarks showed better results by using the `prefetch.global.L1` and `prefetch.local.L1` inline PTX assembly for fetching the data to the L1 caches.

### 2.1.3 Control code simplification

The GPU cores are relatively simple units and their computational efficiency depends on the absence of control code. To increase the instruction level parallelism and avoid possible branch divergence, the implementation adopts three commonly used techniques: i) use of lookup tables, ii) unrolling loops, and iii) branch elimination. The lookup tables are used for selecting the solver and setting the proper values in specific variables. The benefits of loop unrolling are most profound in the preparation of the Rosenbrock solver `ROS` when using the sparse matrix. Finally, limited branch elimination by fusing loops or merging 25  branches also improves the execution time.

### 2.1.4 Decreasing the GPU idle time

The biggest challenge when using accelerators in hybrid HPC architectures is the imbalance created by the uneven workload of tasks. While the application uses GPUs as accelerators, only the CPU cores responsible for communication and handling of GPUs are active, leaving the remaining cores idle. Allocating more processes on the unused CPU core creates more imbalance 30  between the accelerated processes and the CPU-only processes. To address this, the goal is to increase the GPU utilization by assigning more than one GPU per process.

| Node | CPU | Cores | RAM | Accelerators | | | Peak Performance |
|---|---|---|---|---|---|---|---|
| | | # | GB | NVIDIA GPU | Cores | VRAM | GFlops (DP) |
| DELL® dx360 | Xeon X5650 | 12 | 48 | 2× M2070 | 2×448 | 2×6 GB | 1159 (128 CPU + 2×515 GPU) |
| JURECA Node | Xeon E5-2680 v3 | 24 | 128 | 2× K80 | 2×4992 | 4×12 GB | 7568 (3840 CPU + 2×1864 GPU) |
| IBM® S822LC | POWER8 | 20 | 256 | 4× P100 | 4×3584 | 4×16 GB | 21667 (467 CPU + 4×5300 GPU) |

**Table 1.** Hardware configurations used for performance evaluation.

The are two ways to do this: i) over-subscription, and ii) using the *Multi-Process Service (MPS)* (Nvidia, 2015a). The first way is to allow more than two CPU cores (MPI processes) from one node to be offloaded to the accelerators. The downside of this approach is that only one process can access the GPU (exclusive access). Although there will be some benefit compared to the case of using a single process, it is possible that some of the tasks will underutilize the available hardware. The number of GPUs per node and VRAM memory available in each GPU dictates the total number of CPU cores that can run simultaneously. An alternative approach is to use the *MPS* that allows concurrent execution of kernels and memory transfers from different processes on the same node. The latter requires a GPU accelerator with compute capability 3.5 or higher.

## 2.2 Future GPU kernel optimizations

The execution time of individual chemical kinetics tasks depends on the input data of each cell grid. Different task execution times create an imbalance between tasks running on the same GPU. A promising approach to address this challenge is the use of Dynamic Parallelism: the ability for each GPU thread to call other kernels and spawn more GPU threads to increase the available concurrency.

## 3 Results and evaluation

This section presents: (i) the total impact on simulation time, and (ii) the accuracy and correctness of the accelerated model. Three different hardware and software environments were used to measure the impact of the code acceleration (Table 1):

- An iDataPlex dx360 M3 compute node that contains two Intel® Xeon®X5650 6-core processors running at 2.6GHz coupled with two Tesla M-series GPU M2070 accelerators (Fermi architecture). The application is compiled using the Intel compiler (`ifort` ver. 14.0.2) for improved native performance.

- A JURECA (Jülich Research on Exascale Cluster Architectures) computation node that contains two Intel® Xeon®12-core E5-2680 v3 running at 2.5GHz coupled with two Tesla K80 (Kepler architecture) accelerators. The application is compiled using the Intel compiler (`ifort` ver. 16.0.4) for improved native performance. The runtime environment during execution includes `-cpus-per-task=2` to schedule the execution between actual CPU threads and cores.

| Number of columns | 8192 columns with 90 levels |
|---|---|
| Number of grid points | 737280 grid points |
| Number of chemical species | 155 species and 310 chemical reactions |
| Spectral resolution | T42L90MA |
| Ordinary Differential Equations Solver | Ros3: 3-stage, L-stable pair of order 3(2) (Sandu et al., 1997) |

**Table 2.** Experimental configuration of the simulation.

| Configuration | Median CPU exec time (s) | Median Accelerated exec time (s) | Performance over CPU |
|---|---|---|---|
| Intel Xeon X5650 + M2070 | 4.502 | 0.999 | 4.50× |
| Intel Xeon E5-2680 v3 + K80 | 1.476 | 0.283 | 5.21× |
| IBM Power8 + P100 | 3.040 | 0.149 | 20.40× |

**Table 3.** Median execution time and achieved speedup for the non-accelerated (CPU-only) code on a single core and the accelerated version of the kernel for the three platforms. The extracted execution timing results are for the timestep after the initialisation of the simulation.

– An IBM® S822LC compute node equipped with two 10-core 2.92 GHz POWER8 processors (Fluhr et al., 2014; Stuecheli, 2013) with turbo up to 4 Ghz. Simultaneous multithreading is set to 4, for optimum performance in HPC applications. The application is compiled using the IBM compiler (`xlf` ver. 15.1.5). The execution of the `mpirun` command includes `-map-by L2cache -bind-to core:overload-allowed` to reduce the contention of the cache and function units.

To test the model scalability within a compute node, the evaluation uses a representative benchmark simulation with a horizontal resolution of 128 grid points in the longitudinal direction and 64 grid points in the latitudinal direction with 90 vertical levels. Table 2 details the experimental set-up for the results shown in this section. The chemical solver of MECCA in EMAC has a default relative tolerance (`rtol`) of 1E-2 absolute tolerance (`atol`) of 1E1; for key short-lived radicals the `atol = 1` is used. The default E5M2 KPP chemistry batch option is used, along with model namelist set-up `NML_SETUP=E5M2/02b_qctm`, without dynamics nudging and with the diagnostic submodels D14CO, DRADON, S4D, and VISO switched off. The simulation runs with the SCAV submodel disabled to reduce deviations for soluble species whose aqueous-phase chemistry is solved with KPP1 by the submodel SCAV.

## 3.1 Application performance

This section compares the performance of the CPU-only and the GPU-accelerated version of the application. The evaluation uses only one node to avoid any MPI communication interference in the results and limits the period of simulated time to 24 hours. The execution time does not include the initialization of the application and the file I/O operations.

The accelerated version of the kernel achieves an order of magnitude performance improvement over the CPU-only version, as shown in Table 3. We note that the theoretical performance difference between the CPU core and the accelerator is larger

| Configuration | MPI Processes | CPU exec time (s) | Accelerated exec time (s) | Performance over CPU |
|---|---|---|---|---|
| 2× 6-core Intel Xeon X5650 + 2× NVIDIA M2070 | 2 MPI processes | 5199 | 2358 | 2.27× |
| | 12 MPI processes | 1388 | 1368 | 1.01× |
| 2× 12-core Intel E5-2680 v3 + 2× NVIDIA K80 | 4 MPI processes | 7362 | 3384 | 2.17× |
| | 24 MPI processes | 1756 | 1473 | 1.19× |
| 2× 10-core IBM POWER8 + 4× NVIDIA P100 | 4 MPI processes | 2294 | 918 | 2.50× |
| | 20 MPI Processes | 814 | 437 | 1.86× |
| | 40 MPI Processes | 766 | - | 1.75× over 20 MPI Procs |

**Table 4.** Application execution time and achieved speedup of the three node configurations for 24 hours simulated time.

in the P100 platform. The performance improvement of the kernel can provide an indication of the expected application production performance improvement.

Table 4 presents the execution time in seconds and the gain in performance between the GPU-accelerated and CPU-only versions for the two platforms. The node-to-node comparison allows us to compare the accelerated version with the best perfor-
mance achieved using the CPU-only version, using all available cores. The performance gain of the application differs between the available platforms. The M2070 accelerator unit contains only 6GB of memory, limiting the number of processes that can be offloaded in parallel to the accelerator. Moreover, the M2070 accelerator does not support the *Multi-Process Service* (Nvidia, 2015a). Thus, in this experiment, we use the CPU to run the remaining processes instead of accelerators. Despite the significant 2.27× performance gain over execution with two processes, the performance gain over using the entire node is limited. The
accelerated processes complete each timestep before the CPU-only processes, causing severe load imbalance between different MPI processes. Thus, the benefit of having additional accelerators is not reflected in the total attainable performance and an advanced scheduling approach is required between the CPU cores and accelerators.

On the K80 and P100 platforms, the application uses the Multi-Process Service to interact with the accelerators using more than two MPI processes. In the Power8 platform, the best CPU-only performance is achieved using over-subscription to the
CPUs, by using two MPI processes assigned to physical cores. The accelerated version achieves 1.75× performance gain over the best performing CPU-only configuration, clearly showing the benefit of accelerating the chemical kinetics on GPUs. The accelerated version with 40 MPI processes achieves lower performance than the 20 MPI processes (not shown).

The performance gain in ODE systems with varying rate coefficients is greater due to the re-calculation of the coefficients inside the solver. The portion of accelerated workload increases in the case of a non-homogeneous system, resulting in greater
performance gain when GPU accelerators are used, due to the massively parallel architecture. Table 5 shows the results for 24 hours simulated time using the M2070 and K80 accelerators. The achieved performance gain increases from 1.19× to 1.56× when re-calculating `RCONST` at each substep. The changes required for the automatic re-calculation of `RCONST` at each substep are included under the `dev` branch of the source-to-source parser.

| Configuration | Rate coefficients | CPU time | Acc. time | Speed-up |
|---|---|---|---|---|
| 2×(6-core X5650 + M2070) | Constant | 1388 | 1368 | 1.01× |
| | Varying (substep `RCONST`) | 7061 | 5921 | 1.19× |
| 2×(12-core E5-2680 + K80) | Constant | 1756 | 1473 | 1.19× |
| | Varying (substep `RCONST`) | 2600 | 1662 | 1.56× |

**Table 5.** Application execution time in seconds and achieved speedup with/without recalculation of `RCONST` at each KPP substep.

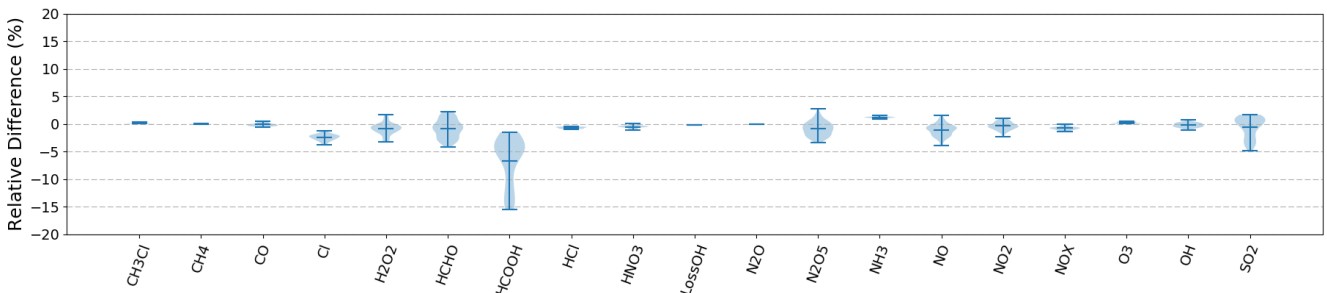

**Figure 3.** Aggregated mass difference between the accelerated and non-accelerated version in the final month after two years of simulation time for different chemical species with the SCAV submodel disabled.

## 3.2 Numerical and model accuracy

To validate the correctness, and evaluate the accuracy of the accelerated implementation we calculate the numerical and modeling difference over two years simulated time. The expected level of relative accuracy (relative numerical difference) for the chemical kinetics calculation is 0.1%, i.e. about 3 accurate digits (Zhang et al., 2011). To calculate the relative difference, we

compare the output of chemical element concentrations between the CPU only and accelerated version after the first time-step. The results show a median difference less than 0.000000001% with the maximum difference value depending on the number of iterations for solving the equations. This is well within the accuracy criterion, asserting the numerical correctness of the GPU kernel.

The variance in floating point results between different architectures is well known and observed in scientific applications

that require high-precision floating point operations (Corden and Kreitzer, 2012; Langlois et al., 2015). The `ifort` compiler produces intermediate results stored with an extended precision that provide greater precision than the basic floating point formats of 64bit. On the other hand, the GPU accelerators don't support the extended precision values, reducing the floating point accuracy in the results. Furthermore, the `REAL(16)` declaration in Fortran programs is implemented in software for the Power8 architecture. Despite the small difference in the results, the model remains stable when running over a two year

simulation time period.

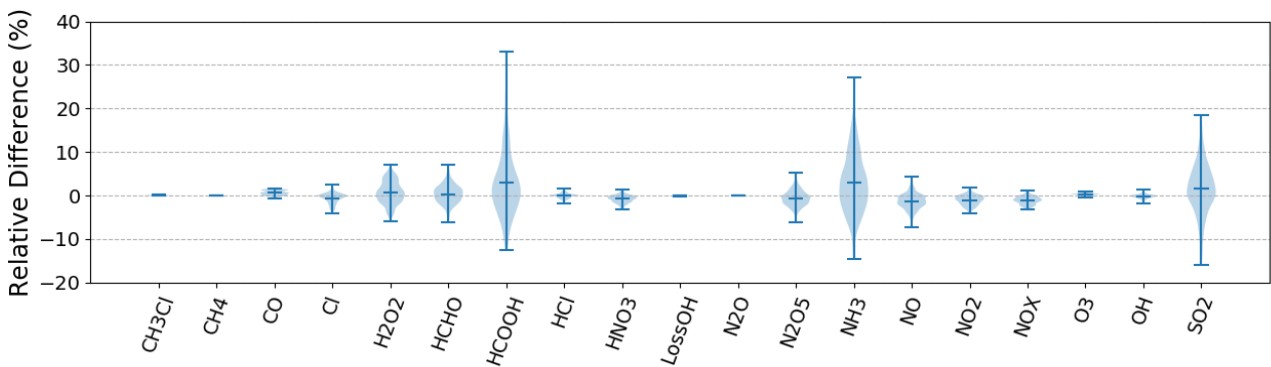

**Figure 4.** Aggregated mass difference between the accelerated and non-accelerated version in the final month after one year of simulation time with the SCAV submodel enabled.

To examine the impact of the inherent data difference on the model accuracy, we compare the results of aggregated mass of chemical species between the CPU-only and GPU-accelerated version, over two years simulated time. Figure 3 presents the chemical species aggregated mass difference distributions between the accelerated and non-accelerated versions. For simplicity, we include 19 important chemical species. The results show that the median value of the difference in aggregated mass is less than 5%, for 18 out of 19 species and higher for HCOOH. The differences are well within the expected margin of differences stemming from architecture and compiler implementations (not specific to GPU). The impact of scavenging amplifies the minor errors created by the different architecture. Figure 4 presents the relative difference after one year simulation with the SCAV (scavenging) module enabled. In this case, median values fall within the 5% limits, with a wider range of extreme values compared to the simulation with the SCAV module disabled. The largest deviations from zero seem to be for soluble species whose aqueous-phase chemistry is very important and solved with KPP1 by the submodel SCAV. The ODE system for aqueous-phase chemistry is notoriously much stiffer than the one for the gas-phase.

Figure 5 compares the output results of the CPU and GPU simulation and their relative difference for zonal mean $O_3$ and surface concentrations of $SO_2, NH_3$ and OH. The relative difference is calculated as the difference over the mean of the two simulations. The largest relative differences appear in areas with the lowest (close to zero) concentrations of chemical elements. The results show the correctness of the accelerated code and its numerical accuracy.

## 4 Conclusions

The global climate model ECHAM/MESSy Atmospheric Chemistry (EMAC) is used to study climate change and air quality scenarios. The EMAC model constits of a nonlocal dynamical part with low scalability and local physical/chemical processes with high scalability. Advancements in hardware architectures over the last three decades have greatly improved not only the spatial and temporal resolution of climate models but also the representation of key processes. The slow adoption

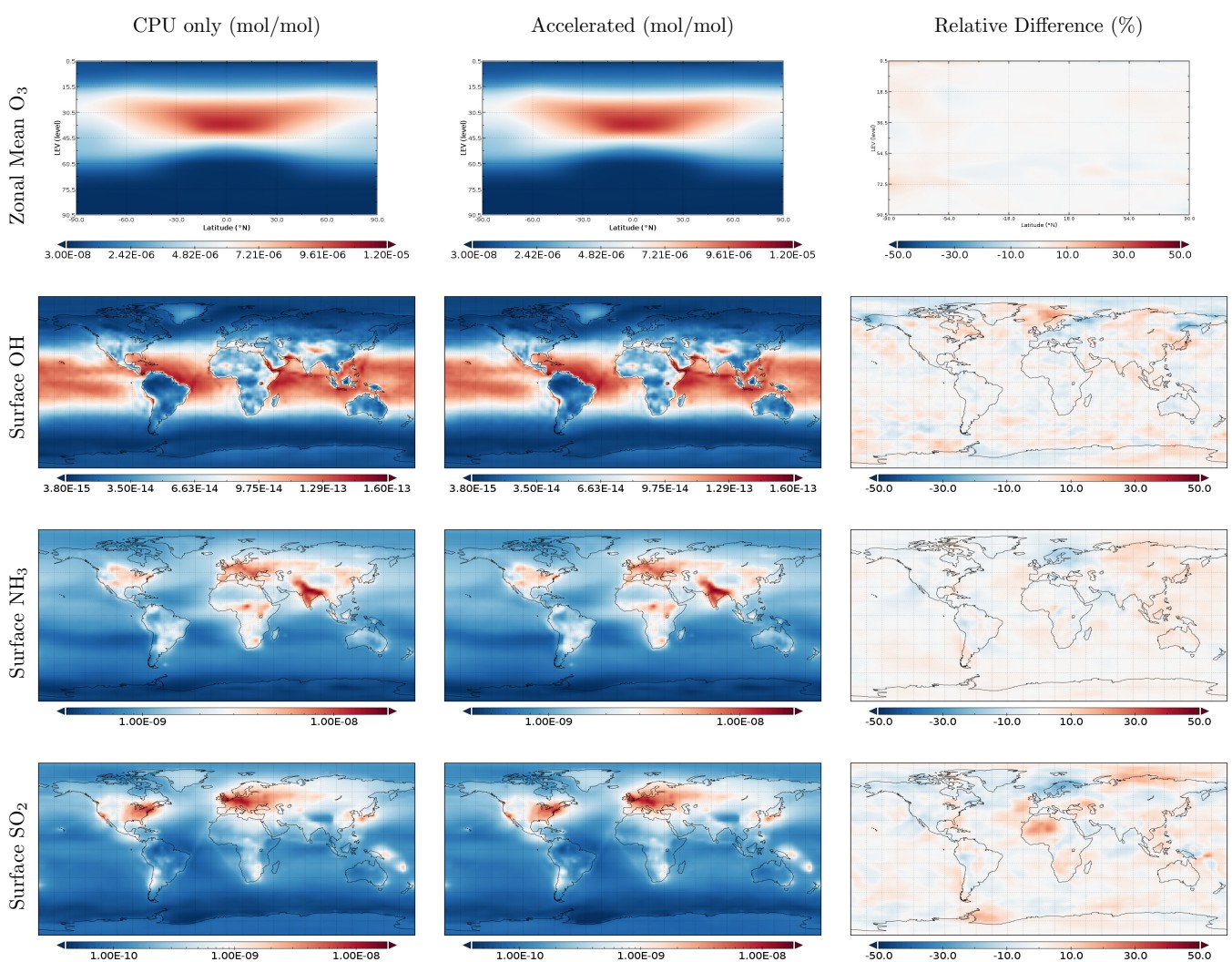

**Figure 5.** Output of CPU-only (1st column), GPU accelerated (2nd column) two-year simulations and their relative difference (3rd column) for different chemical species: $O_3$ zonal mean (1st row), and surface level $OH$, $NH_3$ and $SO_2$ concentrations (2nd, 3rd, and 4th row respectively).

of accelerators in the climate and weather models has positioned the climate simulation community behind other scientific communities (Nvidia, 2017).

In this paper, we presented the acceleration of the KPP chemical kinetics solver in the EMAC chemistry-climate model using a source-to-source parsers to transform the solver to CUDA accelerated code. The parser supports all Rosenbrock family numerical solvers (Ros2, Ros3, Ros4, Rodas3, Rodas4) that are available in the KPP numerical library. A performance evaluation, using two CUDA enabled accelerators shows an achieved speedup of up to $20.4\times$ of the kernel execution time and up to $1.75\times$ node-to-node application performance gain. A comparison of the aggregated global mass of 19 chemical elements between the CPU-only and accelerated version of the application to verify the correctness of the transformation. The numerical accuracy was also assertained, with the relative difference is found to be less than 5% when comparing the output of the accelerated kernel the CPU-only code for 19 chemical elements after two years of simulation time.

The approach followed, including the computational workload division and the developed GPU solver code can potentially be used as the basis for hardware acceleration of numerous geoscientific models that rely on KPP for atmospheric chemical kinetics applications. This work is a marked step forward to increase the resolution of climate simulations using chemical kinetics and expand the usage of GPU accelerators in Earth system modeling.

# 5   Code availability

A consortium of institutions continuously develops the Modular Earth Submodel System (MESSy). The usage of MESSy and access to the source code is licensed to all affiliates of institutions which are members of the MESSy Consortium. Institutions can become a member of the MESSy Consortium by signing the MESSy Memorandum of Understanding. More information can be found on the MESSy Consortium Website (http://www.messy-interface.org).

The FORTRAN to CUDA code-to-code compiler is developed in CUDA and Python and it is included in the EMAC model development release. In addition, the source code is included in public repository under open license, to allow the developer community to contribute (Alvanos and Christoudias, 2017b; The Cyprus Intitute, 2016). It parses the auto-generated MECCA KPP solver code and produces a CUDA library that can be linked to and called directly from within the MESSy FORTRAN code at the time of compilation. In addition, the source code contains a test script to validate the correctness of the transformation and evaluate the performance using the five available Rosenbrock solvers. When executed, the test script runs the following tests: (i) Checks if the parser executes without errors, (ii) checks if the produced file compiles without errors, (iii) executes the application to detect runtime errors and memory violations, and (iv) compares the output of the accelerated version with the output of the serial Fortran version for possible differences of the output.

## Appendix A:  Optimization flags

For completeness we include the optimization flags for the compilers used in the two systems:

| Solver | Base (s) | Occupancy (s) | Prefer L1 (s) | Privatization (s) | Prefetch (s) | Simplification (s) | Total Speedup |
|---|---|---|---|---|---|---|---|
| Ros2 | 9.60 | 9.348 | 9.41 | 6.77 | 6.42 | 6.05 | +58.48% |
| Ros3 | 10.71 | 10.43 | 10.43 | 7.79 | 7.39 | 7.16 | +49.58% |
| Ros4 | 12.54 | 12.28 | 12.24 | 9.52 | 9.18 | 9.05 | +38.65% |
| Rodas3 | 12.58 | 12.31 | 12.28 | 9.51 | 9.17 | 9.01 | +39.59% |
| Rodas4 | 17.67 | 17.31 | 17.27 | 14.16 | 14.06 | 14.10 | +25.33% |
| Mean Speedup | - | +2.21% ± 0.3 | +0.22% ± 0.43 | +29.18% ± 6.35 | +3.69% ± 1.93 | +2.58% ± 2.59 | +19.14% ± 5.12 |

**Table 6.** Execution time in seconds and speedup running the kernel on a M2070 CUDA enabled accelerator placed in a node equipped with two Intel Xeon X5650 processors at 2.67 GHz with synthetic input.

– For Intel platform, we used the ifort compiler with the following optimization flags: `-O3 -g -debug full -traceback -fp-model precise -fp-model source -fp-speculation=safe -fpp -align all`

– For IBM PowerPC platform, we used the xlf compiler version 15.1.5 with the following optimization flags: `-O3 -g -qnohot -qarch=auto -qcache=auto -qsource -qtune=auto -qsimd=auto -qinline=auto:level=10 -qprefetch=aggressive:dscr=7`

## Appendix B:  Impact of optimizations

The evaluation uses a microbenchmark based on the accelerated kernel to demonstrate the effectiveness of the code transformations. The microbenchmark uses the input concentrations extracted from the execution of one time-step of the application. We use five different variations of the Rosenbrock solver and two GPU accelerators, the M2070 and the P100, and we record only the GPU execution time. Tables 6, 7, and 8 of the evaluation present the results of the kernels for each group of optimizations applied.

The *occupancy* optimization and the change of the cache organization (*Prefer L1*) have minor performance gains. Furthermore, the performance gain of these optimizations in the P100 accelerators is limited within the margin error. The most notable negative impact (-18%) in the performance is the usage of the *Prefetch* intrinsics inside the source code in the P100 accelerator. The newest accelerators contain advanced hardware prefetchers that decrease the impact of the memory optimization. Thus, this forced us to modify the source code for the P100 platform by disabling the software prefetcher.

On the other hand, the *Privatization* of global arrays using shared memory, registers, and stack allocated arrays, gave the biggest benefit by reducing the off-chip memory traffic. The performance gain for the memory optimizations is less pronounced in the newest CUDA enabled architectures, as they provide a better memory subsystem compared with the M2070 accelerator. The simplification of the source code has the greatest impact in the newest architectures due to reduced control dependencies inside the accelerated code.

| Solver | Base (s) | Occupancy (s) | Prefer L1 (s) | Privatization (s) | Prefetch (s) | Simplification (s) | Total Speedup |
|---|---|---|---|---|---|---|---|
| Ros2 | 3.77 | 4.09 | 4.48 | 3.20 | 2.94 | 2.96 | 22.40% |
| Ros3 | 4.21 | 4.61 | 4.55 | 3.71 | 3.71 | 3.43 | 19.11% |
| Ros4 | 5.06 | 5.50 | 5.46 | 4.58 | 4.29 | 4.27 | 16.50% |
| Rodas3 | 5.09 | 5.52 | 5.45 | 4.57 | 4.29 | 4.26 | 17.12% |
| Rodas4 | 7.41 | 8.05 | 7.94 | 6.93 | 6.61 | 6.58 | 12.09% |
| Mean Speedup | - | -7.9% ± 0.39 | 1.25% ± 0.21 | 15.12% ± 2.77 | 3.15% ± 1.81 | 0.21% ± 0.19 | +17.12% ± 3.78 |

**Table 7.** Results running the kernel on a K80 CUDA enabled accelerator hosted in a node equipped with an Xeon E5-2680 v3 at 2.5 GHz.

| Solver | Base (s) | Occupancy (s) | Prefer L1 (s) | Privatization (s) | Prefetch* (s) | Simplification (s) | Total Speedup |
|---|---|---|---|---|---|---|---|
| Ros2 | 1.23 | 1.26 | 1.18 | 0.98 | 1.20 | 0.90 | 37.24% |
| Ros3 | 1.39 | 1.39 | 1.39 | 1.14 | 1.36 | 1.04 | 34.17% |
| Ros4 | 1.75 | 1.74 | 1.74 | 1.44 | 1.70 | 1.29 | 40.79% |
| Rodas3 | 1.74 | 1.75 | 1.76 | 1.39 | 1.70 | 1.23 | 41.23% |
| Rodas4 | 2.75 | 2.74 | 2.75 | 2.12 | 2.66 | 1.95 | 40.57% |
| Mean Speedup | - | +0.06% ± 1.09 | -0.14% ± 2.81 | +22.48% ± 3.87 | -18.14% ± 2.03 | +9.62% ± 3.30 | +40.57% ± 3.04 |

**Table 8.** Results running the kernel on a P100 CUDA enabled accelerator hosted in a node equipped with two Power8 processors. In the P100 platform, we removed the prefetch optimization, as it decreases the performance in all cases.

*Disclaimer.* Any opinions, findings and conclusions or recommendations expressed in this material are those of the author and do not necessarily reflect the views of the funding agencies.

*Acknowledgements.* The research leading to these results has received funding from the European Community's Seventh Framework Programme (FP7/2007-2013) under Grant Agreement No 287530 and from the European Union's Horizon 2020 research and innovation programme under grant agreements No 675121 and No 676629. This work was supported by the Cy-Tera Project, which is co-funded by the European Regional Development Fund and the Republic of Cyprus through the Research Promotion Foundation.

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
