# Peer review of "GPU accelerated atmospheric chemical kinetics in the ECHAM/MESSy (EMAC) Earth system model (version 2.52)"

_Geoscientific Model Development, 2017_

## Referee Comment (RC1) · Anonymous Referee #1 · 21 May 2017

The manuscript presents a description and accuracy and performance evaluation of a software that enables the established Earth System Model EMAC to offload its most demanding calculations onto GPUs. Such a development is very welcome and highly needed in order to address the challenges of atmospheric chemistry modelling under changing climate. The manuscript is well written and structured. However, results for non-homogeneous ODE system are not presented and the evaluation of model accuracy should be expanded both in terms of key atmospheric species and of time integration length.

[Figure]

Major comments

1) At page 6 it is written that RCONST, the array of the rate constants, is calculated only once at the beginning of the integration. This is correct for the setup the authors have chosen. However, the default MECCA chemical mechanism in EMAC 2.52 is translated by KPP in a ODE system with varying (rate) coefficients which has to be integrated as a non-homogeneous system. This means that the RCONST array has to be re-calculated at each substeps. How does this additional memory allocation affect accuracy and performance ? The test should be done with the unchanged gas.eqn file of MECCA without replacement of the reactions for MIM1.

2) To Figure 3. The largest deviations from zero seem to be for soluble species whose aqueous-phase chemistry is very important and solved with KPP1 by the submodel SCAV. The ODE system for aqueous-phase chemistry is notoriously much stiffer than the one for the gas-phase. This hints to a potential amplification of errors. One could check the accuracy of a simulation without scavenging.

3) For climatic simulations it is important to ensure no drift in the results. A 10-year simulation is desirable when one thinks about chemistry-climate feedbacks in the stratosphere and methane lifetime in the troposphere.

4) The manuscript would profit from showing the distribution of the differences at simulations' end. For instance, the relative differences for the zonal mean ozone and surface OH are worth showing. Given their importance for aerosols differences for surface NH3 and SO2 are desirable.

Minor comments

1) A reference to the actual KPP version used (likely v 2.2.3-rs) published in Sander et al.(2011) (http://www.geosci-model-dev.net/4/373) and slightly modified from KPP-2.1 by Sandu and Sander (2006) ( http://www.atmos-chem-phys.net/6/187/2006/) is better. The main difference to KPP1 is the generated code in Fortran 90 and not anymore in

[Figure]

FORTRAN.

2) What is meant by statistical reduction of results at page 6? Why is it needed?

3) Page 13, line 1. How can it be "climatic" if the climate is understood to be at least a ~30 yr average state of the atmosphere?

4) To Figure 3. Could you show the differences for OH, N2O5, HNO3, HCOOH and H2O2? Could you also explain the meaning of the bar widths and shaded areas?

5) Which chemical species shows the maximum error of 0.54% ?

6) The chemical solver of MECCA in EMAC has a default relative tolerance (rtol) of 1E-2 and a absolute tolerance (atol) of 1E1 although for key short-lived radicals the atol = 1 is used. This implies a maximum 1% error.
* * *

---

## Referee Comment (RC2) · Anonymous Referee #2 · 12 Jun 2017

The authors present the implementation and evaluation w.r.t. performance and accuracy of a source-to-source translation tool for KPP kernels in the EMAC earth system model. CUDA kernels and corresponding calling code is generated by parsing Fortran code generated by KPP. A representative benchmark is evaluated on 2 different platforms using 2 different generations of Nvidia GPUs. The performance impact of various optimizations is studied using microbenchmarks. Furthermore, the tool is released open source under a permissive MIT license and the specific release used to produce the manuscript is deposited on zenodo with a DOI, both of which are commendable.

**1 Remarks / questions**

1. Why did you decide for CUDA in favour of OpenCL (which is also mentioned on p. 4)?

2. Have you considered directive based approaches e.g. OpenACC?

3. Why is it necessary to parse the code generated by KPP? Could the integration be done at an earlier stage e.g. AST, before actually generating code and parsing it again? (p. 5)

4. Have you considered using an existing source-to-source translation framework e.g. the ROSE compiler framework, which does have a CUDA backend? (p. 5)

5. Did you consider other ways of subdividing the computation than columns? (p. 5)

6. Could you have refactored the solver to split matrices among threads such that they fit in on-chip memory? (p. 8)

7. Where does the 2.5x speedup figure in Table 4 come from? And where is there a 1.75x speed of 40 vs. 20 MPI processes?

**2 Minor comments**

1. What are the scalability limitations of the EMAC model? (p. 2)

2. Does KPP generate *either* Fortran or C code (equivalent) or *both* Fortran and C code calling each other? (p. 3)

3. What do you mean by "the linear algebra can be solved off-line before the execution of the application"? (p. 4)

4. "the design architecture for memory allocation is subject to change when the impact of coalesced memory access is more prominent" (p. 6): Do you mean with future GPU generations? Do you not expect memory access limitations to be lifted further?

5. Why is more memory required for Pascal? (p. 7)

6. Lower occupancy results in a reduced number of concurrent thread blocks per SM (p. 8)

7. Do you mean thread local variables rather than scalar variables for privatization? (p. 9)

8. The description of the strategy for reducing load imbalance (p. 9) is not very clear: what do you hope to gain by "offloading" more processes than available GPUs?

9. Can yo elaborate how "representative benchmark" was chosen? (p. 10)

10. What do you mean by "the performance gain of these optimizations in the P100 accelerators is limited within the margin error"? (p. 14)

**3   Grammar / wording (page / line)**

1/14: to the CPU-only code

2/28: executes the gas phase chemical kinetics independently because these are independent of

3/1: in terms of portability

3/4: KPP provides ... analysis routines

3/15: execution times

4/7: parallel instructions

4/25: assigns an equal number of

5/24: GPU threads

Algorithm 2: Step 2: Call kernels

7/5: for future GPU architectures if applicable

7/6: to the GPU

7/15: the limiting factor for memory allocation

7/16: at least

8/5: on the GPU

8/11: remove "the occupancy of"

8/15: Launch Bound

8/21: remove "width of"

9/8: computational efficiency

9/30: for each GPU thread

10/14: scalability within a compute node

11/7: remove "with"

13/10: EMAC model consists of

15/3: a better memory subsystem

---

## Author Response (AR1)

**" GPU accelerated atmospheric chemical kinetics in the ECHAM/MESSy (EMAC) Earth system model (version 2.52)" Reply to Referee Comments**

Michail Alvanos, Theodoros Christoudias
m.alvanos@cyi.ac.cy, christoudias@cyi.ac.cy

July 6, 2017

We would like to thank the referees for their careful reading of our manuscript. We are most grateful for the comments, constructive criticism and very useful suggestions received on how to improve the paper.

Please find our detailed answers to the comments and a new version of the paper, which we hope satisfactorily address the points raised during the discussion.

**1 Referee 1**

**1.1 Major comments:**

*At page 6 it is written that RCONST, the array of the rate constants, is calculated only once at the beginning of the integration. This is correct for the setup the authors have chosen. However, the default MECCA chemical mechanism in EMAC 2.52 is translated by KPP in a ODE system with varying (rate) coefficients which has to be integrated as a non-homogeneous system. This means that the RCONST array has to be re-calculated at each substeps. How does this additional memory allocation affect accuracy and performance ?*

The developed source-to-source parser automatically recognizes the requirement for the additional re-calculation of the rate coefficients and injects the necessary calls into the produced CUDA code. The re-calculation increases the overall computational workload and execution time significantly. The portion of accelerated workload increases in the case of a non-homogeneous system, resulting in greater performance gain when GPU accelerators are used, due to the massively parallel architecture. This is now added to the text.

| Configuration | Rate coefficients | CPU time | Acc. time | Speed-up |
|---|---|---|---|---|
| 2×(6-core X5650 + M2070) | Constant | 1270 | 1260 | 1.01× |
| | Varying (substep `RCONST`) | 7061 | 5921 | 1.19× |
| 2×(12-core E5-2680 + K80) | Constant | 1756 | 1473 | 1.19× |
| | Varying (substep `RCONST`) | 2600 | 1662 | 1.56× |

Table 1: Application execution time in seconds and achieved speedup with/without recalculation of `RCONST` at each KPP substep.

Table 1 shows the results for 24 hours simulated time. Using the K80 accelerators the achieved performance gain increases from 1.19× to 1.56× when re-calculating `RCONST` at each substep. The changes required for the automatic re-calculation of `RCONST` at each substep are included in the source-to-source parser.

*The largest deviations from zero (Figure 3) seem to be for soluble species whose aqueous-phase chemistry is very important and solved with KPP1 by the submodel SCAV. For climatic simulations it is important to ensure no drift in the results. A 10-year simulation is desirable when one thinks about chemistry-climate feedbacks in the stratosphere and methane lifetime in the troposphere.*

Given the limited computational resources available to us, we re-ran the simulation for 2 years, disabling the SCAV submodel to acquire better comparison results for chemical concentrations whose aqueous-phase chemistry is solved with KPP. All the results and the relevant text in the manuscript were updated accordingly.

*The manuscript would profit from showing the distribution of the differences at simulations end. For instance, the relative differences for the zonal mean ozone and surface OH are worth showing. Given their importance for aerosols differences for surface NH3 and SO2 are desirable.*

Fig. 2 presents the output results of the simulation and the relative difference. The results show the correctness of the accelerated code and its impact in the numerical accuracy. The largest relative differences appear in areas with the lowest concentration (close to zero) of chemical elements.

**1.2 Minor:**

*Slightly modified from KPP-2.1 by Sandu and Sander (2006) is better.*

We added the additional citation of KPP.

*Page 13, line 1. How can it be "climatic" if the climate is understood to be at least a 30 yr average state of the atmosphere?*

The time restrictions and available computation resources limit the simulation to two years. The goal of the paper is not present a new climatic model and verify it, but rather to improve the performance. Moreover, the climatic comparison in the valuation section proves the correctness of the transformation and not the correctness of the model.

*To Figure 3. Could you show the differences for OH, N2O5, HNO3, HCOOH and H2O2?*

Figure 1 now also presents the differences for the additional chemical elements. The text was updated accordingly.

*Which chemical species shows the maximum error of 0.54% ?*

The elements HNO3, NO3, and the temporary LossJ3103a have the greater difference, especially when the solver needs multiple steps to solve the equations. We removed the reference of 0.54% in the final version of the paper because the relative difference depends on the number of solver substeps.

[Figure]

Figure 1: Output of CPU only (1st column), GPU accelerated (2nd column) and their relative difference (3rd column) for chemical species: O$_3$ zonal mean, and surface level OH, NH$_3$ and SO$_2$ concentrations (2nd,3rd, and 4th row respectively).

*The chemical solver of MECCA in EMAC has a default relative tolerance (rtol) of 1E-2 and a absolute tolerance (atol) of 1E1 although for key short-lived radicals the atol = 1 is used.*

The chemical solver of MECCA in EMAC has a default relative tolerance (rtol) of 1E-2 and a absolute tolerance (atol) of 1E1 although for key short-lived radicals the atol = 1 is used, as the reviewer indicated.

[Figure]

Figure 2: Aggregated mass difference between the accelerated and non-accelerated version of one year simulation for different time-steps with 19 chemical species.

**2    Referee 2**

*Why did you decide for CUDA in favour of OpenCL?*

Our currently available hardware infrastructure utilizes CUDA enabled accelerators. The vendor (NVidia) provides strong debugging support and improved performance when using the CUDA compiler framework compared with the OpenCL solution.

*Have you considered directive based approaches e.g. OpenACC?*

Yes, we have considered these approaches. We found that is would still be necessary to modify the source code in order to successfully run the application and gain substantial performance benefits. In our view, OpenACC could be helpful in (relatively) simpler cases that do not require significant source code refactoring. In our case, the complexity of the solver and automatic generation of the code limits the applicability of the directives usage.

*Why is it necessary to parse the code generated by KPP? Could the integration be done at an earlier stage e.g. AST, before actually generating code and parsing it again?*

Yes, someone can modify the KPP to produce the accelerated kernel. There is already a commercial version of KPP (Kppa: the Kinetic preprocessor accelerated) that can produce this code. However, our application uses a modified version of KPP, with additional pre-processing components (known as KP4) necessitating a source-to-source parser.

*Have you considered using an existing source-to-source translation framework e.g. the ROSE compiler framework, which does have a CUDA backend? (p. 5)*

Using compiler frameworks would improve the compatibility of the source code. We avoided the requirement of additional compiling of multiple files and dependencies on other packages. Our goal was to provide a simple utility for making the transformation without requiring the distribution of additional software packages with a different license.

*Did you consider other ways of subdividing the computation than columns? (p. 5)*

During the development of the application, we considered a more fine-grained approach of parallelizing parts of the solver. Unfortunately, (i) the parallelization potential is limited due to branching, and (ii) the overhead of issuing individual tasks and transferring the data outweighs the benefit of accelerating the code. In the future, we aim for further improvement of

the accelerated parts by extracting additional parallelism through invoking kernels from within already accelerated parts of the code (e.g. using CUDA Dynamic Parallelism).

*Could you have refactored the solver to split matrices among threads such that they fit in on-chip memory? (p. 8)*

Using the grid-point calculation as a computation task, it is not possible to break the matrices due to cell inter-dependencies. Using more fine-grained parallelism (e.g. subdividing parts of the solver) than the present granularity can decrease the memory requirements, however, the amount of available parallelism is ultimately limited by data and control dependencies. Thus, our implementation using one thread for calculation of chemical kinetics for each grid-point, and grid-point offloaded in user-defined arrays of columns provides the best performance.

*Where does the 2.5x speedup figure in Table 4 come from? And where is there a 1.75x speed of 40 vs. 20 MPI processes?*

Indeed, there was an error in the reported values of GPU execution time. The execution time for 4 MPI processes is 918 seconds and not 1652. We corrected the value in the text. The execution time for the best CPU-only configuration is 766 seconds when using 40 processes, and for GPU is 437 seconds when using 20 processes. The resulting performance gain comparing the best configurations is $1.75\times$ as reported.

**2.1 Minor:**

*What are the scalability limitations of the EMAC model? (p. 2)*

The scalability limit of the application is the communication. The application scales well until few hundred cores and then the performance does not improve with higher core counts.

*Does KPP generate \*either\* Fortran or C code (equivalent) or \*both\* Fortran and C code calling each other? (p. 3)*

KPP produces either Fortran or C code (equivalent).

*What do you mean by "the linear algebra can be solved off-line before the execution of the application"? (p. 4)*

In a typical chemical mechanism, the pattern of chemical interactions leads to a sparse Jacobian (with the majority of entries equal to zero). The KPP uses sparse linear algebra to reduce the execution time. The sparsity structure depends only on the chemical network and not on the values of concentrations or rate coefficients. Thus, the lookup tables are constant, allowing to KPP and KP4 to unroll the loops and remove the indirect memory accesses. We added the clarification in the manuscript.

*"the design architecture for memory allocation is subject to change when the impact of coalesced memory access is more prominent" (p. 6): Do you mean with future GPU generations? Do you not expect memory access limitations to be lifted further?*

Indeed, with future GPU generations, we expect the memory access limitation to be lifted further. During our experience with the application development, we noticed that a large number of registers causes spillage to stack memory. Thus, the impact of memory coalescing

Table 2: Available GPU architectures.

| Name | M2070 | K80 | P100 | V100 |
|---|---|---|---|---|
| Architecture | Fermi | Kepler | Pascal | Volta |
| Shared Mem/L1 Cache | 64 KB | 64 KB | - | 128 KB |
| Dedicated Shared Memory | - | - | 64KB | - |
| Dedicated L1 Cache | - | - | - | - |
| L1 / Rd-only cache | - | - | 24 KB | - |
| Rd-only cache | - | 48 KB | - | - |
| L2 Cache Size | 768 KB | 1536 KB | 4096 KB | 6144 KB |

is limited due to stack occupancy by each CUDA thread. If the number of available registers increases or the local memory becomes larger, then the performance gain from coalescing memory access can be significant. The trend in the GPU architectures is the increase of the on-chip memory. For example, Table 3 presents the timeline of CUDA-enabled architectures. The memory hierarchy and the size of caches evolve with each architecture iteration. Thus, we can expect further improvements of the memory subsystem that may require refactoring of the code to achieve the best performance. We added the clarification in the manuscript.

*Why is more memory required for Pascal? (p. 7)*

We believe the reason for additional allocation memory is because the driver allocates additional memory for buffers and caches inside the GPU. There is also a possibility of a software bug in the driver of the accelerator. The Pascal architecture supports the Unified Virtual Memory model that automatically swaps the data between CPU and GPU. To support this feature, the driver allocates additional memory for data structures we transfer to GPU. Thus, it is possible the driver allocated the memory even if we don't use the memory. We added in the text additional information regarding the memory requirements.

*Grammar / wording*

We addressed all the comments regarding the grammar errors.

**3   Changes in the manuscript**

1. Following the suggestion of the first reviewer we have run for a longer period of two years to test the model stability and accuracy, disabling the SCAV submodel to reduce deviations for soluble species whose aqueous-phase chemistry is solved with KPP1. The computational resources available to us did not allow for a full decadal run.

2. We added in the paper comparison of of zonal mean and surface concentrations and their relative differences for important species as suggested by the 1st anonymous referee.

3. We added information on the performance impact of having to recalculate reaction constants at each solver substep.

4. We added additional details regarding the experimental setup of the environment.

5. We added an additional computation platfom, JURECA computing node, equipped with two K80 accelerators and we evaluate the produced accelerated code.

---

## Author Response (AR2)

**" GPU accelerated atmospheric chemical kinetics in the ECHAM/MESSy (EMAC) Earth system model (version 2.52)" Reply to Referee Comments**

Michail Alvanos, Theodoros Christoudias
`m.alvanos@cyi.ac.cy, christoudias@cyi.ac.cy`

August 30, 2017

We would like to thank the referees for their careful reading of our manuscript. We are most grateful for the comments and very useful suggestions received on how to improve the paper.

Following the suggestion of the reviewer we added in the paper comparison of the results with the SCAV (scavenging) module disabled (Figure 3) and enabled (Figure 4). We modified the text to reflect the differences and the impact of the module in comparison with and without acceleration.